# Spatiotemporal Pattern Analysis of China's Cities Based on High-Resolution Imagery from 2000 to 2015

**Hanchao Zhang [1] , Xiaogang Ning [2], Zhenfeng Shao [3],* and Hao Wang [2,4]**

1    School of Remote Sensing and Information Engineering, Wuhan University, 129 Luoyu Road,
     Wuhan 430079, China; zhc_geo@whu.edu.cn
2    Institute of Photogrametry and Remote Sensing, Chinese Academy of Surveying and Mapping,
     Beijing 100830, China; ningxg@casm.ac.cn (X.N.); wanghao@casm.ac.cn (H.W.)
3    State Key Laboratory of Information Engineering in Surveying, Mapping and Remote Sensing,
     Wuhan University, 129 Luoyu Road, Wuhan 430079, China
4    Beijing Key Laboratory of Urban Spatial Information Engineering, Beijing 100038, China
*    Correspondence: shaozhenfeng@whu.edu.cn; Tel.: +86-27-6877-9859

**Abstract:** The urbanization level in China has increased rapidly since beginning of the 21st century, and the monitoring and analysis of urban expansion has become a popular topic in geoscience applications. However, problems, such as inconsistent concepts and extraction standards, low precision, and poor comparability, existing in urban monitoring may lead to wrong conclusions. This study selects 337 cities at the prefecture level and above in China as research subjects and uses high-resolution images and geographic information data in a semi-automatic extraction method to identify urban areas in 2000, 2005, 2010, and 2015. City size distribution patterns, urban expansion regional characteristics, and expansion types are analyzed. Results show that Chinese cities maintained a high-speed growth trend from 2000 to 2015, with the total area increasing by 115.79%. The overall scale of a city continues to expand, and the system becomes increasingly complex. The urban system is more balanced than the ideal Zipf distribution, but it also exhibited different characteristics in 2005. Urban areas are mostly concentrated in the eastern and central regions, and the difference between the east and the west is considerable. However, cities in the western region continuously expand. Beijing, Shanghai, Tianjin, and Guangzhou are the four largest cities in China. Approximately 73.30% of the cities are expanding in an extended manner; the urban form tends to be scattered, and land use efficiency is low. The new urban areas mainly come from cultivated land and ecological land.

**Keywords:** urban area; urban expansion; spatiotemporal pattern analysis; high-resolution imagery; China

## 1. Introduction

The urban population is increasing with the development of urbanization, along with the rapid expansion of urban areas [1]. In particular, China's urbanization entered a new stage from 2000 onwards. The 19th National Congress of the Communist Party of China, the 13th Five-Year Plan, the National New Urbanization Plan (2014–2020), and the Central City Work Conference (2015) proposed to rationally control urban development boundaries, protect basic farmlands, and optimize cities [2,3]. The internal space structure promotes compact and intensive urban development and efficient green development to solve the blind expansion of cities, faster land urbanization than population urbanization, extensive and inefficient land construction, urbanization and industrialization mismatch, urban spatial distribution, and unreasonable scale structures [4]. Many problems, such as insufficient resources and environmental carrying capacity, exist in addition to outstanding ecological and environmental problems [5,6]. The reasonable control of urban growth boundaries, optimization of

urban internal spatial structures, and promotion of compact urban development have become inherent requirements for sustainable urban development [7,8]. The monitoring and analysis of urbanization play extremely important roles in accurately capturing the urbanization process, scientifically implementing urban planning and smart city construction [9], and promoting urban sustainable development.

Urbanization can be measured in terms of population urbanization and land urbanization [10]. Population urbanization can be expressed by the urbanization rate (the ratio of the urban population to the total population) [11]. Land urbanization can be seen in terms of non-agricultural urban areas [12]. Compared with the intangible indicator of population transfer caused by population urbanization, land urbanization, i.e., urban spatial expansion, is the direct result of urbanization acting on geospatial space [13]. Hence, it is easier to measure objectively. Remote sensing technology exhibits the advantages of providing real, objective, and current information and requires a low cost; it has become an important technical means for monitoring land urbanization [14–17]. The primary task of urbanization monitoring is to determine urban areas or boundaries. However, urban areas and boundaries have many related concepts. These definitions have different starting points and characteristics [18,19]. Many studies have regarded construction land or impervious surfaces in land use classification results as urban areas [20,21]. Some researchers have defined an urban area by considering the functions of urban land [22]. Numerous sources of image data are used in monitoring urbanization [23]. During the early days, Moderate Resolution Imaging Spectroradiometer (MODIS) images were the primary source of data for monitoring urbanization [24,25]. Subsequently, images from medium-resolution satellites, such as the Landsat series [26,27], were widely used [28–33]. As the number of launched high-resolution satellites increases, many researchers have attempted to extract urban areas from high-resolution remote sensing images [34–40]. However, these images are mostly used on a single city scale due to their ability to automate extraction.

Many scholars have analyzed urban expansion patterns and characteristics by obtaining urban areas [41]. The urban spatial pattern is the overall spatial pattern of a city, the spatial distribution pattern of various elements within a city, and the comprehensive performance of the dynamic evolution process. The overall spatial pattern of a city is based on the urban scope. The internal elements of a city are based on land use [42] and land cover [43], including the spatial characteristics of scale, shape, density, distribution, location, and relationship. Many scholars have also used multisource data, such as remote sensing images. Research on urban overall spatial patterns from static expressions and dynamic analyses are in multiple scales, such as land parcels, blocks, functional areas, cities, urban agglomerations, regions, countries, and the world. Li uses the rank-size distribution and the Gini coefficient method to analyze the urban spatial pattern of the Yangtze River Economic Belt based on nighttime stable light data, multi-temporal urban land products, and multiple sources of geographic data [44]. Zheng analyzes the urban spatial pattern of Hubei province by 'point-axis system' theory, light index model, gravity model and social network analysis. Liu makes a high-resolution multi-temporal mapping of global urban land using Landsat images based on the Google Earth Engine Platform and Normalized Urban Areas Composite Index [45].

On the basis of urban area extraction, research on urban spatial form introduces fractal dimension, compactness, aggregation index, urban center of gravity transfer index, spatial autocorrelation index, maximum patch index, expansion speed, expansion intensity, extended contribution rate, and landscape [46–48]. Expansion indices and equal measure indicators have been used, and superposition analysis, convex hull analysis, fan analysis, quadrant quantile analysis, buffer analysis, circle analysis, center of gravity migration analysis, and other methods have been conducted to analyze the extent, pattern, and direction of urban boundaries and expansions [49–51]. Subsequently, studies on urban overall spatial morphological characteristics and spatiotemporal evolution have been conducted.

The research of various scholars provides numerous methods for understanding the spatiotemporal expansion process of China's urbanization. However, several shortcomings remain. First, each study presents a different understanding of urban concepts, inconsistent extraction standards, and considerable differences in monitoring results, thereby causing difficulty in performing accuracy



verification and comparative analysis. Second, the results are mostly based on MODIS and Landsat series satellite imagery [52]. The low resolution of these images leads to the poor accuracy of urban extraction results. The beginning of the 21st century was the period of rapid development in Chinese cities; however, high-resolution remote sensing images were still scarce at that time [48]. No set of high-precision monitoring products was available nationwide.

To solve the aforementioned problems, a semi-automatic method of urban boundary extraction was proposed by using high-resolution image and geographic information data. Urban landscape and form characteristics, geographical knowledge, and a series of standardized rules were combined to generate a high-precision and consistent urban boundary.

The current study uses a semi-automatic method to extract urban area using high-resolution remote sensing imagery and geographic information data. Urban landscape and form characteristics, geographical knowledge, and a series of standardized rules were combined to generate high-precision and consistent urban areas of 337 cities in 2000, 2005, 2010, and 2015. Then, a city size distribution model was verified, urban expansion regional characteristics were analyzed at three main levels, i.e., the city, province and region levels, and expansion types were also analyzed.

## 2. Study Areas and Data

### 2.1. Study Areas

A total of 337 cities (including capital cities, prefecture-level cities, autonomous prefectures, regions, and alliances, but excluding Hong Kong, Macao, and Taiwan) are selected as research areas (Figure 1). A municipal district is selected as the urban extraction scope, and a city with administrative division adjustment is subjected to the 2018 municipal jurisdiction. There are 4 megacities (population of 10 million people or more), 13 supercities (population above 5 million but below 10 million), 36 large cities (population above 1 million but below 5 million) and hundreds of small-medium cities (population below 1 million). The research areas are divided into four regions based on their economic characteristics: the eastern, central, western, and northeastern regions. Cities at the prefecture level and above include national economic, political, and population centers, and their urbanization level can represent the highest in the region. During the early 21st century, cities at the prefecture level and above progressed rapidly in terms of economic and social development, and urban areas expanded rapidly. These cities are representatives of the monitoring and analysis of urbanization spatial expansion in China.

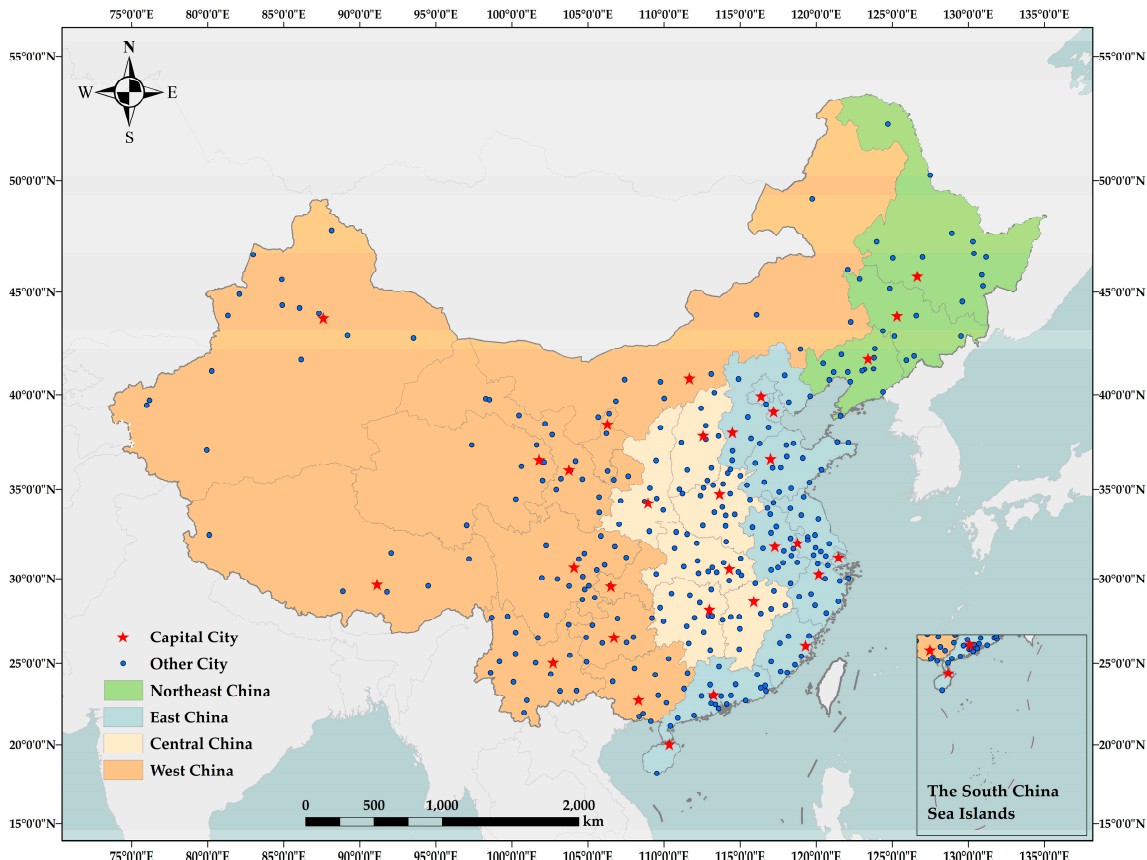

**Figure 1.** Distribution of 337 cities and four regions across the country.

## 2.2. Data

This study uses high-resolution remote sensing images as the primary source of data, which cover the entire study area. The data types listed in Table 1 are arranged in accordance with the adopted order of priority. Among these data, the 2015 remote sensing images of the study area is primarily from the "Geographical Conditions Monitoring (GCM)" national project [53], which is obtained from the National Basic Geographic Information Center. Remote sensing images of 2010 were mostly provided by the Ministry of Natural Resources. Insufficient images were supplemented by collecting the first national census image data of the country. Satellite imagery was obtained from September to December, and the resolution is generally higher than 1m. In 2000 and 2005, remote sensing images, including high-resolution remote sensing images and aerial photographs, were mostly from surveying and mapping geographic information departments. Insufficient images were supplemented through network downloading and purchasing, and images from September to December were preferred. A small number of areas, below 1%, were supplemented by images from the years before and after due to the wide coverage of the study area and the long-time span. To ensure the consistency of the extraction results, the image data were uniformly resampled to a resolution of 2 m. The auxiliary data primarily included the results of the GCM and basic geographic information, which were obtained from the Ministry of Natural Resources. These data included land cover, urban road elements, geographical names, townships, and county administrative divisions and can provide good references for urban area extraction.

**Table 1.** Primary remote sensing image data used in this study.

| Year | Month | Data Type/Source (Resolution) | Coverage Ratio |
|------|-------|-------------------------------|----------------|
| 2000 | April, July, August, September, October, November, December | Aerial Images (1 m), IKONOS (1 m), small number of Landsat data (30 m) | WorldView-1/2 (22.37%), aerial images (19.76%), IKONOS (14.62%), others (43.25%) |
| 2005 | June, July, August, September, October, November, December | QuickBird (0.61 m), aerial images (1 m), IKONOS (1 m), SPOT 5 (2.5 m), minor Landsat data (30 m) | |
| 2010 | September, October, November, December | WorldView-1/2 (0.5 m), QuickBird (0.61 m), aerial images (0.5 m), SPOT 5 (2.5 m), ALOS (2.5 m), CIRS-P5 (2.2 m) | |
| 2015 | September, October, November, December | WorldView-1/2 (0.5 m), Pleiades (0.5 m), aerial images (0.5 m), SPOT 6/7 (1.5 m), Mapping Satellite I (2 m), ZY-3 (2.1 m) | |

## 3. Methods

The method adopted in this study includes three major steps: (1) data preprocessing, (2) urban area extraction, and (3) urban spatiotemporal pattern analysis. Figure 2 shows the overall analysis process of the spatiotemporal pattern of Chinese cities. First, high-resolution remote sensing images obtained in 2000, 2005, 2010, and 2015 were screened. Second, image preprocessing, fusion, mosaicking, and cropping were performed on the data to satisfy the requirements. Third, the method proposed in [35] was used to extract urban areas, which were then manually modified in accordance with city characteristics and the extraction principle adopted to obtain the final urban area. Lastly, the spatiotemporal pattern of Chinese cities was analyzed from three aspects: urban size distribution pattern, urban expansion area characteristics, and extended form.

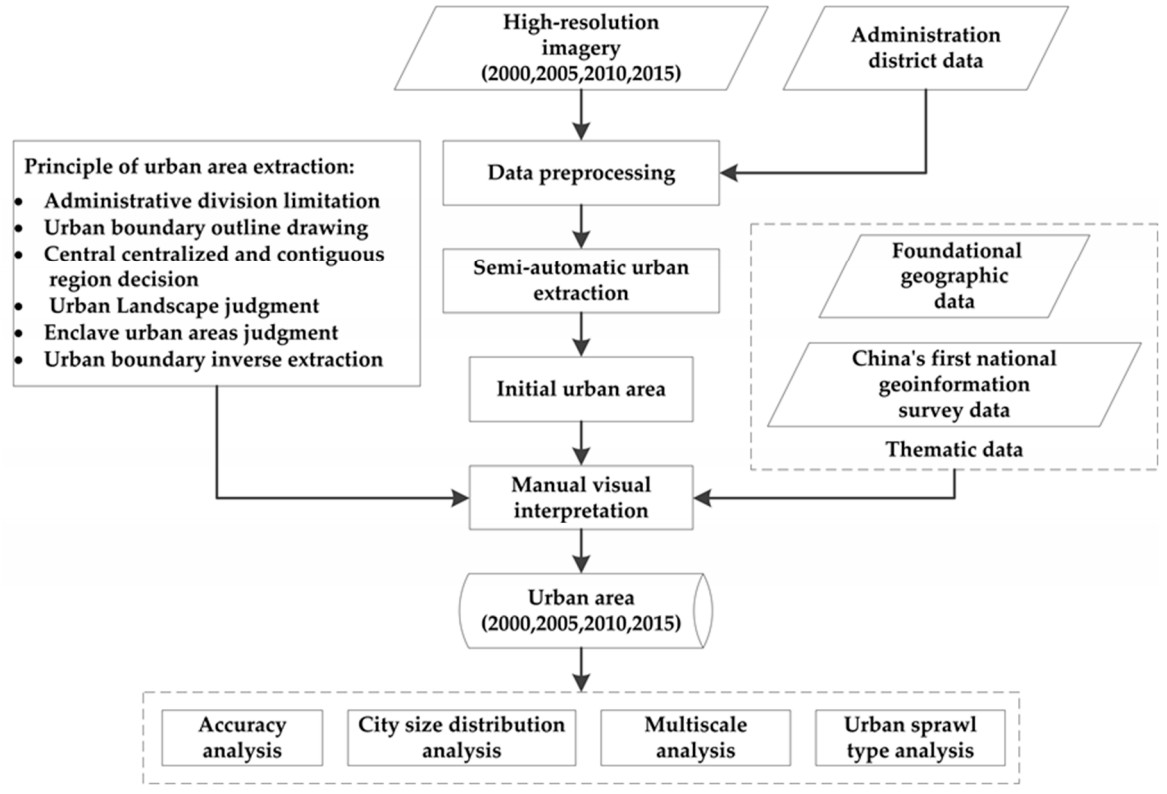

**Figure 2.** Flowchart of the analysis of China's urban expansion characteristics.

### 3.1. Data Preprocessing

Data preprocessing was accomplished using professional software, such as ArcGis10.3, ERDAS 2016, and ENVI 5.3.

(1) Data screening

Data screening requires selecting images with a resolution higher than 2 m within a city's jurisdiction. The image phase is preferably summer, and the cloud snow coverage is preferably less than 10%. The 2000 national coordinate system is used, and other coordinate system images are required after correcting the registration. In addition, some data from the Ministry of Natural Resources require extensive inspection and screening due to problems including the inability to open data copying errors, incomplete data, and redundant data.

(2) Orthorectification correction

This method primarily includes the orthorectification of a single image and the registration of images from different coordinate systems. Orthorectification corrects imaging deviations caused by the imaging method used by the sensor during image capture, the influence of a change in the external orientation element, terrain fluctuation, Earth's curvature, atmospheric refraction, and the influence of Earth's rotation. Image registration completes the geometric transformation of images from other coordinate systems to the 2000 national coordinate system by selecting the same name point.

(3) Image mosaicking

This process includes image stitching and acquiring uniformity. After image registration is completed, all the images are uniformly resampled to a resolution of 2 m in order to mosaic. A large number of remote sensing images are spliced to obtain a geometrically uniform color image by searching for the optimal mosaic edges and applying image tonal adjustment.

(4) Image cropping

This process is primarily used in urban administrative division to appropriately cut an entire remote sensing image obtained by splicing, thereby removing unnecessary parts and reducing the amount of data.

(5) Administrative division boundary treatment

Administrative division line vector data are obtained from the Ministry of Natural Resources, and the administrative division line vector of a city jurisdiction is selected by comparing the city jurisdiction code in the National Administrative Division Manual. The vector data of the administrative divisions of each municipal district are merged to generate the administrative boundaries of a city's jurisdiction. The demarcation line between the sub-city jurisdiction and the administrative division of the city is reserved.

### 3.2. Urban Area Extraction

Urban area extraction is a combination of automatic computer extraction and manual interpretation correction. The automatic extraction of residential polygons from high-resolution remote sensing images is performed using fused right-angled and right-angled features [38]. The polygons of the settlements are superimposed with the location of the district government, whereas the polygons of the concentrated contiguous settlements, where the district government is located, are used as the initial urban area. The classification system and standards for geographic country monitoring indicate that the original urban area should be manually interpreted and modified in accordance with the urban area extraction principles and rules to obtain the final urban area.

The automatic method uses feature-level-based fusion of right-angle-corners and right-angle-sides to extract the human settlements. It is composed of five steps, namely (1) detection of line segments, (2) detection of Harris corners, (3) verification of corners by line segments, (4) construction of built-up area index, (5) thresholding of human-settlement index.

After the automatic extraction of human settlements, manual interpretation is used for urban extraction. Urban area extraction should start from actual city construction completion under full consideration of urban landscape. Buildings are the core; natural landscapes are the auxiliary and roads

are the bond of the urban landscape. Using top-down process, rough initial boundary is extracted first, and then refine it under the principles based on the GCM data, fundamental infrastructure of geographical information. The process should be in reverse chronological order and must ensure the accurate topology between multi-phase boundaries.

The final urban area includes the central city and the enclave-type urban area. The revised principles and rules for an initial urban area are as follows:

(1) Principle for defining the boundaries of administrative divisions. An urban area must be within the scope of the administrative division of the city and must not exceed the boundaries of the administrative division.

(2) Principle for drawing outlines of urban area. An urban area outline is preferentially drawn along the boundary of linear objects, such as roads and rivers, and cannot cross buildings (districts) and structures. The inner side of an urban boundary line should have no cultivated land. When an urban boundary falls on a river bank and the surrounding area of the river bank is a green landscape for urban construction land, the urban boundary line is drawn along the high-water line of the river, and the green part is classified as an urban area; otherwise, the road near the river bank is drawn. Urban boundary is preferentially sketched along the border of complete block, which has obvious boundary, such as courtyards or fences. If the block boundary is not obvious or the proportion of unconstructed area in the whole block is high, urban boundary is sketched along the actual boundary of the constructed land.

(3) Principle for determining the central city. The central centralized and contiguous region, which is named as central urban areas, is connected to district government through the road, and the border of the region extends towards outside until the distance of constructed land to central urban areas is more than 50 m.

(4) Principle for determining urban landscapes. An urban landscape mostly includes urban housing construction areas, urban roads, urban green spaces, city squares, parking lots, stadiums, and other landscapes. The block surrounded by urban roads is the most important urban landscape feature.

(5) Principle for judging enclave urban areas. An enclave-type urban area, i.e., an enclave-type concentrated contiguous area, is an important part of an urban area. It exhibits the following characteristics: close contact with the central city through a road; urban landscape features; an area larger than 60 hectares; and contains at least one of the following geographic units (urban-level street offices, administrative departments, industrial and mining areas, development districts, research institutes, universities, and colleges).

(6) Principle for the reverse extraction of urban boundaries from new time to old time. First, the urban area extraction of the latest phase is completed. Second, the inverse incremental updating technique is used to obtain the urban area change element in the order of the current situation from new to old. Lastly, the post-phase area is merged with the urban area change to obtain the previous time. The urban area of the phase ensures accurate topology between multi-period boundaries.

The combination of automatic computer extraction and manual interpretation correction considerably saves computational efficiency compared with performing manual visual inspection and manual sketching to obtain the initial urban area. After testing, the efficiency of automatic identification is determined to be nearly twice higher than that of manual visual interpretation. Combined with the initial urban area interpretation correction, the comprehensive improvement efficiency reaches 30%.

Examples of typical features within urban areas which can help us to identify the urban area in this study are presented in Figure 3.

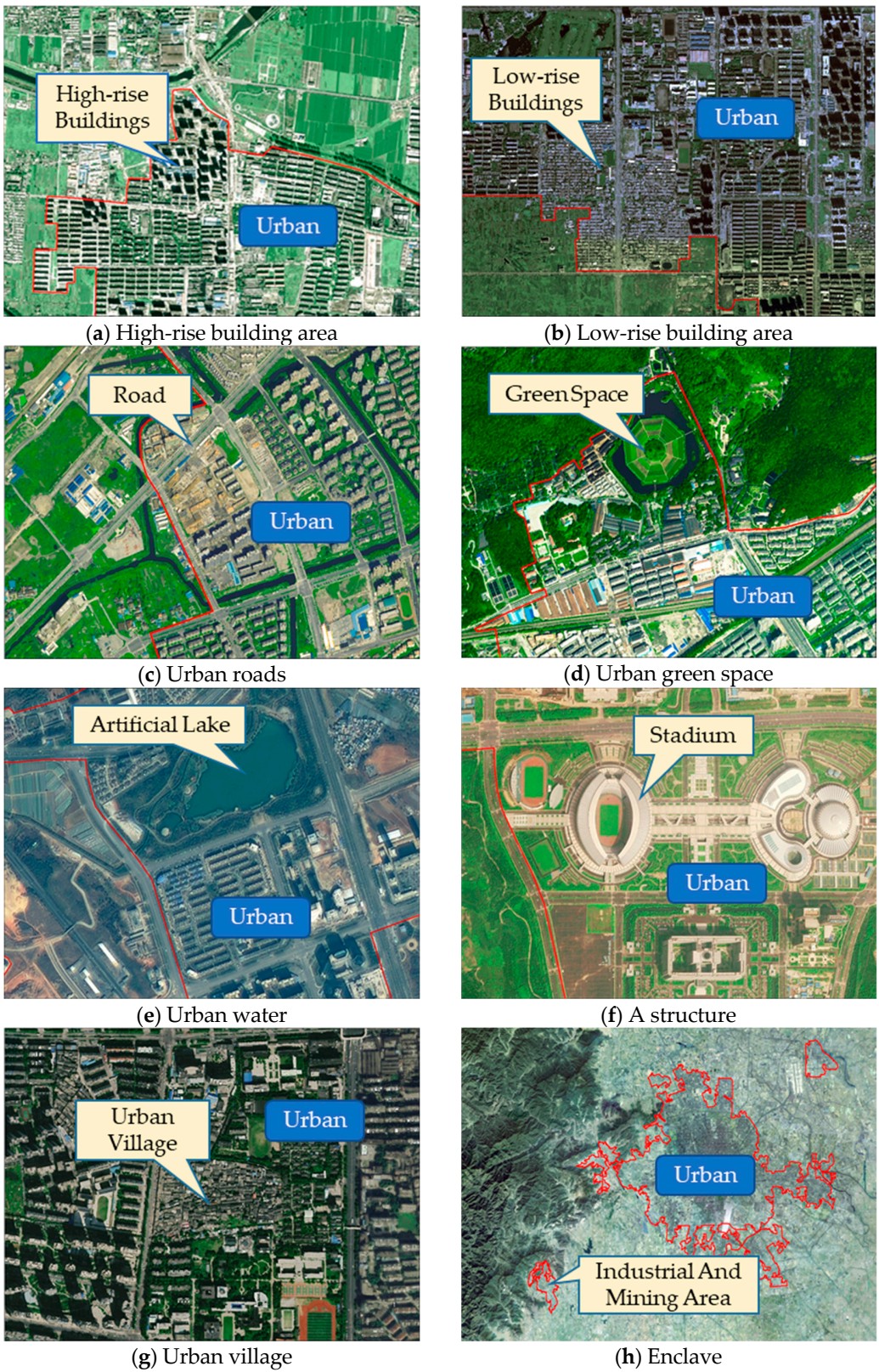

**Figure 3.** Examples of urban feature interpretation symbol: (**a**) an example of high-rise buildings in urban areas; (**b**) an example of low-rise buildings in urban area; (**c**) an example of roads in urban area; (**d**) an example of green space near the edge of urban area; (**e**) an example of urban water, namely an artificial lake, in urban area; (**f**) an example of structure, namely a stadium, in urban area; (**g**) an example of urban village; (**h**) an example of enclave, namely an industrial and mining area, of urban area.

### 3.3. City Size Distribution

City size distribution refers to the hierarchical distribution of cities in a country or region. Urban population, urban area, economic indicators, and composite indicators are frequently used as indicators of urban scale [54,55]. An urban area is an urban scale indicator [56]. The characteristics of urban distribution within a region are determined by analyzing the relationship between the sequence of large and small cities in the region and its scale [57]. The order-scale rule examines the size distribution of an urban system from the relationship between city size and city size order. Its formula is as follows:

$$P_i = P_1 \cdot R_i^{-q} \tag{1}$$

where $R_i$ is the city sorted by size from large to small $i$ with the order $P$, $i$ is the city size with the order $R_i$, $P_1$ is the theoretical value of the first city size, and parameter $q$ is typically called the Zipf index.

In the study of urban systems, the order-scale rule is frequently combined with fractal theory to better understand the characteristics of city size distribution. The fractal dimension ($D$) of city size distribution and q in Equation (1) exhibit the following relationship in accordance with fractal theory:

$$D \times q = R^2 \tag{2}$$

where $R^2$ represents the coefficient of determination, $D$ is the fractal dimension and $q$ is the Zipf index which can reflect the equilibrium of an urban system.

(1) When $q$ is large and $D$ is small, the city size distribution in a region is concentrated, large cities with high ranks are prominent, and small and medium-sized cities with moderate and low ranks are insufficiently developed. The scale difference between cities is considerable, and the urban scale system is unbalanced.

(2) When $D$ is large and $q$ are small, the city size distribution in the region is relatively scattered, the scale of high-level cities is not prominent, and small and medium-sized cities are relatively developed. The city size distribution only differs slightly, and the urban scale system is more balanced.

(3) When $q$ and $D$ are close to 1, the ratio of the size of the first city to that of the smallest city is close to the total number of cities in the region. The city size distribution is close to the ideal state of Zipf, and the proportion of cities in each scale is reasonable.

The theory of city size distribution indicates that a regular sequence distribution is obtained when $q = 1$, namely, the Czech Republic model. In the first distribution, i.e., when $q > 1$, the urban population is concentrated, the urban system is dominated by large cities, and small and medium-sized cities are insufficiently developed. In a sequence distribution, i.e., when $q < 1$, the urban population is scattered, large cities in the urban system are not prominent, and small and medium-sized cities are developed. When $q = 0$, all urban populations are equal and belong to the average distribution. However, when $q = \infty$, the region has only one city.

$lgP_1$ can stand for structural capacity (Sc). For urban systems, when the structural capacity is large, the urban system is complex and the overall scale is large. When the structural capacity is small, the urban system is simple and the overall size is small.

### 3.4. Urban Expansion Analysis Indicator

After obtaining the urban area, urban time–space expansion is analyzed using urban expansion speed and intensity.

(1) Expansion speed $V_i$ is the annual growth rate of an urban area in a certain city within a particular period. It indicates the absolute (area) difference of the expansion speed of different urban areas per unit time.

$$V_i = \Delta U_{ij} / \left( \Delta t_j \right) \times 100\% \tag{3}$$

where $V_i$ is the urban expansion speed, $\Delta U_{ij}$ is the $i$-th research unit urban expansion area in the $j$ period, and $\Delta t$ is the time span of the $j$ period.

(2) Expansion intensity $N_i$ is the annual expansion ratio of an urban area in a certain city relative to the base period within a particular period. It indicates the relative (proportional) difference in the speed of expansion of different urban areas per unit time.

$$N_i = \Delta U_{ij}/\left(\Delta t_j \times M_i\right) \times 100\% \tag{4}$$

where $N_i$ is the urban expansion intensity, $\Delta U_{ij}$ is the $i$-th urban expansion study area in the $j$ period, $\Delta t_j$ is the time span of the $j$ period, and $M_j$ is the total $i$-th unit urban area during the initial stage of the $j$ period.

### 3.5. Urban Expansion Type

The type of urban space expansion typically includes infilling and edge-expansion [58], which can be determined via convex hull analysis [59]. The convex hull of a city's land outline refers to the smallest convex polygon that contains the outline of the city's periphery. The convex hull can be considered an urban area or the potential control area of a city.

The convex shell analysis method can be used to identify the spatial expansion pattern of urban land. If urban land expansion belongs to the filling pattern, then the extended land is mostly inside the convex hull of the urban area. If urban land expansion belongs to the expansion pattern, then the extended land is mostly outside the convex hull of the urban area. That is, in urban land expansion, if the area inside the convex shell is larger than the area outside the convex shell, then the urban land expansion belongs to the filling pattern and vice versa. Figure 4 shows us the examples of two types.

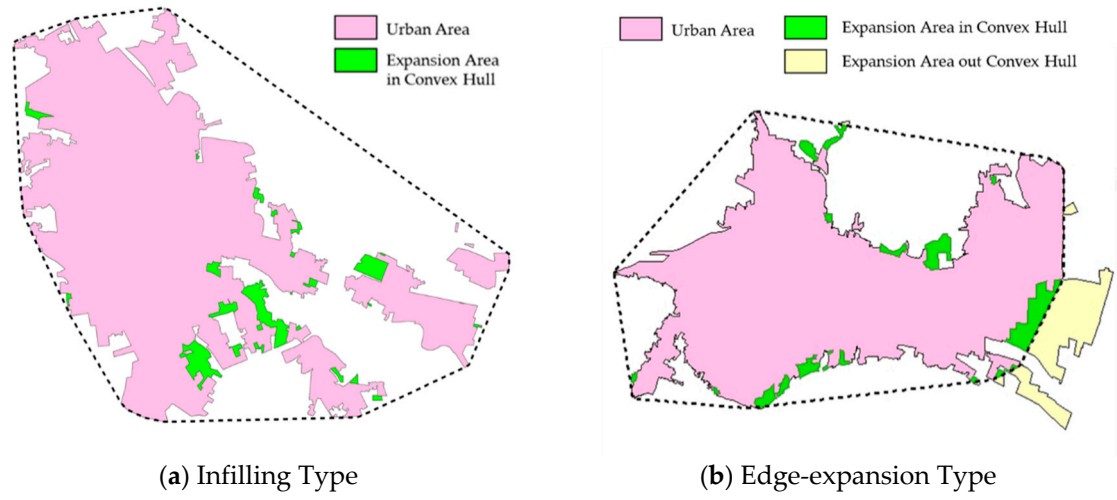

(**a**) Infilling Type　　　　　　　　　　　　　　　(**b**) Edge-expansion Type

**Figure 4.** Examples for urban expansion types.

## 4. Results and Discussion

### 4.1. Accuracy Analysis of Urban Area Extraction Results

This study extracts the urban areas of 337 cities at prefecture level and above. Reliability and high precision are primarily reflected in the following three aspects.

(1) The technical method guarantees high precision of the extraction result. The urban area is first extracted by adopting the automatic extraction method of residential points with right-angled features to ensure the consistency of the extraction results. In addition, the adopted extraction principles and standards are strict and standardized, and operability is strong. Interpreters must strictly follow the six standardized extraction principles and the specific technical regulations so as to ensure the reliability of the urban area extraction results.

(2) As the primary data, high-resolution remote sensing images guarantee the high precision of the extraction result. The adopted remote sensing image exhibits high resolution and can effectively

avoid misclassification and leakage. The results of some studies show that resolution considerably influences the accuracy of urban area extraction. Compared with MODIS and Landsat images, high-resolution images can effectively avoid the misclassification and leakage of urban boundaries and can combine urban and urban–rural areas and townships. Thus, the construction land can be separated using high-resolution remote sensing images, which is difficult to be separated using MODIS or Landsat images.

(3) High-quality auxiliary data guarantee the high precision of extraction results. Difficult areas have real and reliable auxiliary data support to ensure the accuracy of the extracted results. The largest difficulty in extracting urban boundaries is the determination of concentrated contiguous urban landscapes and enclave-type urban areas. The current study solves the three aforementioned urban-related problems by considering the results of GCM and the data of urban functional units, development zones, bonded areas, and urban roads in the 2015 basic geographical condition monitoring results. The large features are identified.

The capital cities are selected to be compared with other results, such as the result of MODIS in the report "East Asia's Changing Urban Landscape" (product A) and the result of manual extraction by Landsat images (product B) in [60]. Two figures from [60] may give us a more succinct comparison of different results (Figures 5 and 6).

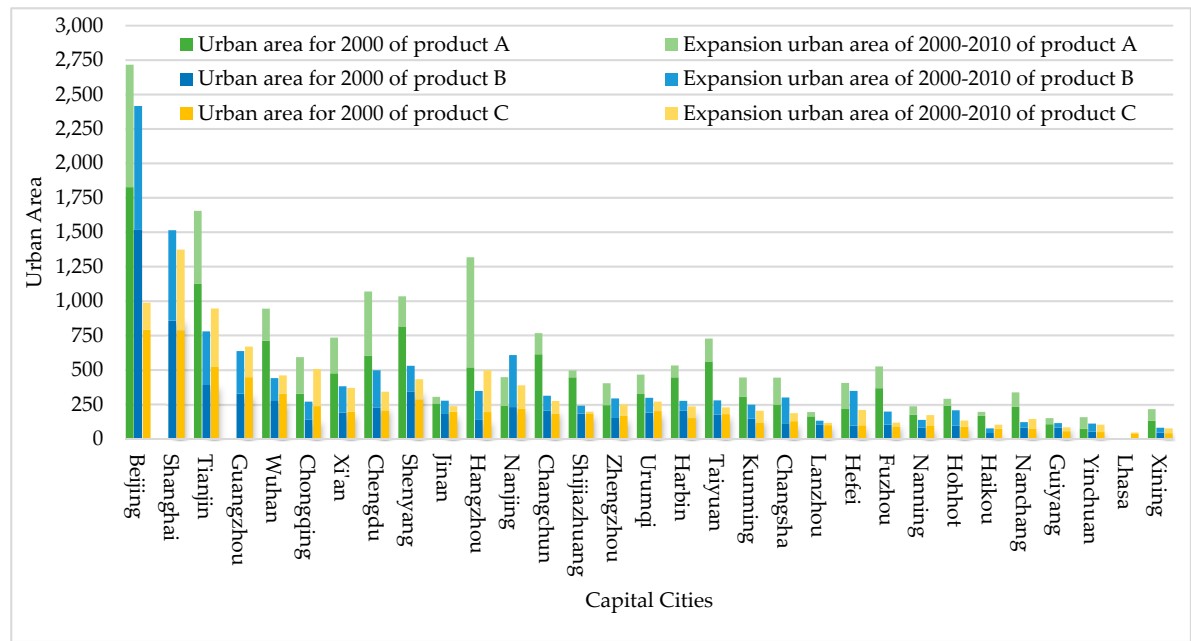

**Figure 5.** The comparison of extraction results of urban areas. Product C is the result of proposed method.

Figure 5 shows us that the area of product A is apparently larger than PRODUCT B and product C. And product B is slightly larger than product A. The t-test tells us that product B and product C are basically consistent. However, product A is quite different from product B and product C. Then, we choose the large different cities to compare.

The comparison of Beijing (Figure 6) shows us that the urban product A is much larger than product B and product C. From the high-resolution images, we can find that there are many mistakes in product A. Many farmlands, forests even some mountains are mistaken for urban areas. The results of product B and the results of this paper are similar. The figures from Figure 6b to Figure 6i show that the urban areas with some high-rise buildings and obvious aggregation in Figure 6b are excluded from product B. However, the low-rise buildings in the urban–rural integration area (Figure 6c), part of the rural area in Figure 6d, and the large farmland areas in Figure 6e were incorrectly classified as built-up areas in the results for B, and the results were compared with the corresponding Landsat5

TM images Figure 6f,g. It can be seen from Figure 6h,i that in these areas, except for Figure 6f, it is obvious that there is a missing phenomenon in product B, and that it is difficult to accurately identify the boundary in Figure 6g–i. Nearby features can easily lead to misunderstandings. According to the extraction results of Beijing, the accuracy of result A is the worst, and the precision of result B is high. The accuracy of our result is the highest. Product A has the worst accuracy, mainly because the resolution of MODIS is much lower than that of the others; it is difficult to achieve the high-resolution extraction result and it no longer joins the comparison of Chongqing city.

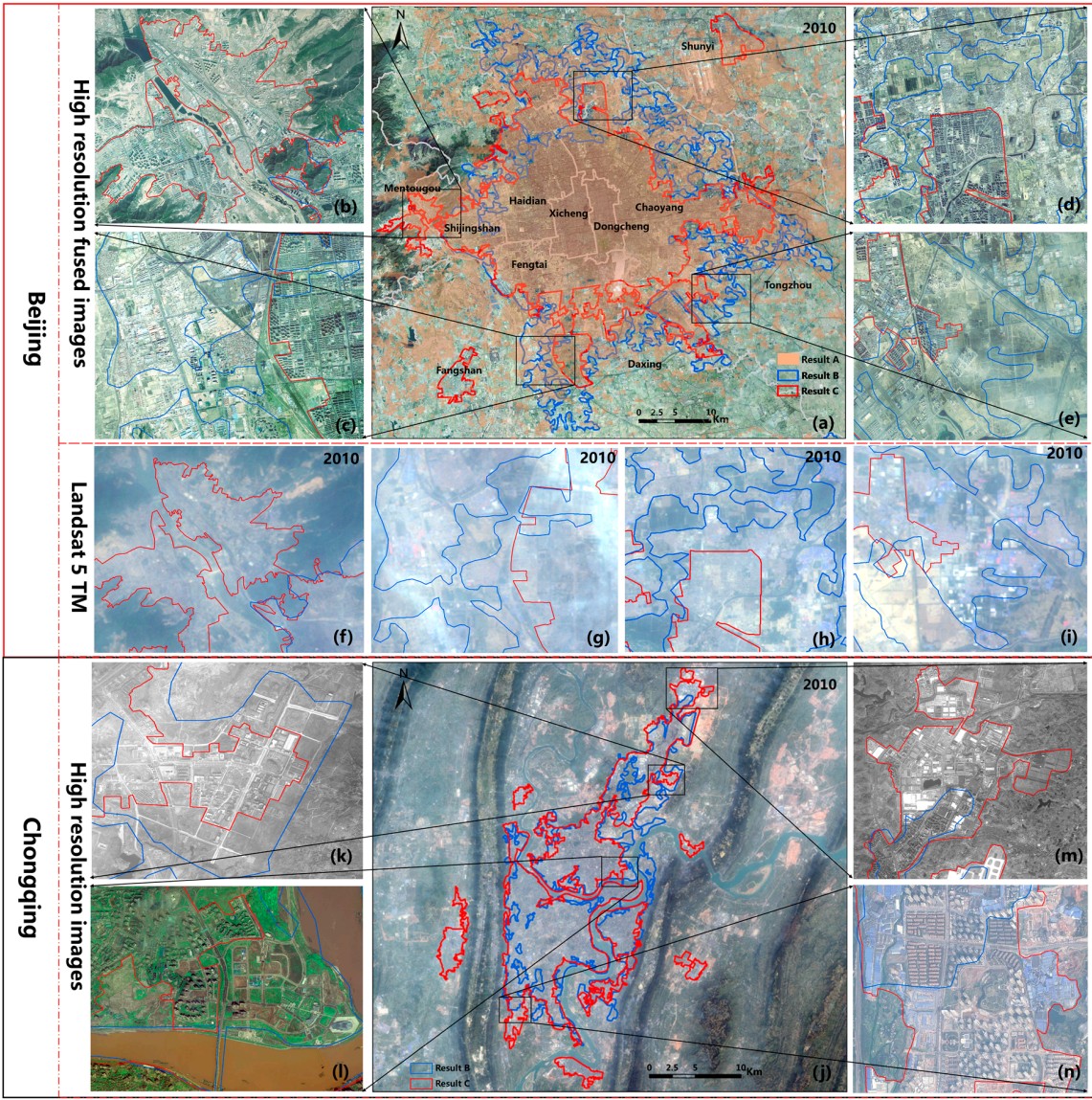

**Figure 6.** Comparison examples for urban boundaries of A, B and C in 2010—Beijing, Chongqing.

In Chongqing, Figure 6j shows the overall situation of the results of the two major urban areas. It can be seen that the extraction results are generally consistent, and the details of specific boundaries are slightly different. Figure 6k–n show the boundary of the more specific differences, from Figure 6k,n showed that there is some leakage in product B; Figure 6l shows that there are some mistakes of farmland and bare land in product B. Figure 6m shows that the results in this paper are more accurate than those of product B.

By comparing the results of the three different data sources extraction, it can be seen that the resolution of the data source for urban area extraction has a remarkable effect on the result accuracy.



Using high-resolution data to extract urban areas can effectively help in avoiding mistakes, omission and ambiguous boundaries; it can also accurately distinguish the city, the integration of urban and rural areas, rural, and accurately obtain the city internal elements distribution, morphology and structure information; these extraction results can play an important role in urban development.

### 4.2. Analysis of City Size Distribution

The total area of Chinese cities was 16,012.84 km$^2$ in 2000, 21,200.41 km$^2$ in 2005, 27,778.11 km$^2$ in 2010, and 34,554.31 km$^2$ in 2015, thereby indicating an increase of 32.40%, 31.04%, and 24.39%. The area in 2015 is 2.16 times that which existed in 2000. This result shows that China's urban growth has maintained a significant growth trend. The urban sequence-scale analysis of the extraction results of the four phases (2000, 2005, 2010, and 2015) is performed by regarding the urban area of China's prefecture level as the city scale index.

After abandoning the unreliable tail of the ranks [61], the results in Figure 7 show that the model fitting coefficient R$^2$ of each period is above 0.97, which indicates that the model fitting degree is better. Moreover, the order-scale method can better describe the distribution of Chinese city size. The scale of the first cities in each period is smaller than the theoretical value. The scale of the top cities is relatively close, thereby indicating that the first degree of China's urban system is insufficiently prominent, and thus, not a typical first distribution. The values of the Zipf index in 2000, 2005, 2010, and 2015 are 0.82, 0.86, 0.85, and 0.82, respectively. The fractal dimension D values are 1.19, 1.13, 1.15, and 1.18, respectively, thereby indicating that the scale distribution of Chinese cities is better than the ideal Zipf distribution. In terms of decentralization, the scale of high-level cities is insufficiently prominent, and small and medium-sized cities are relatively developed. The distribution of urban scales is relatively small, and the scale system of cities is more balanced.

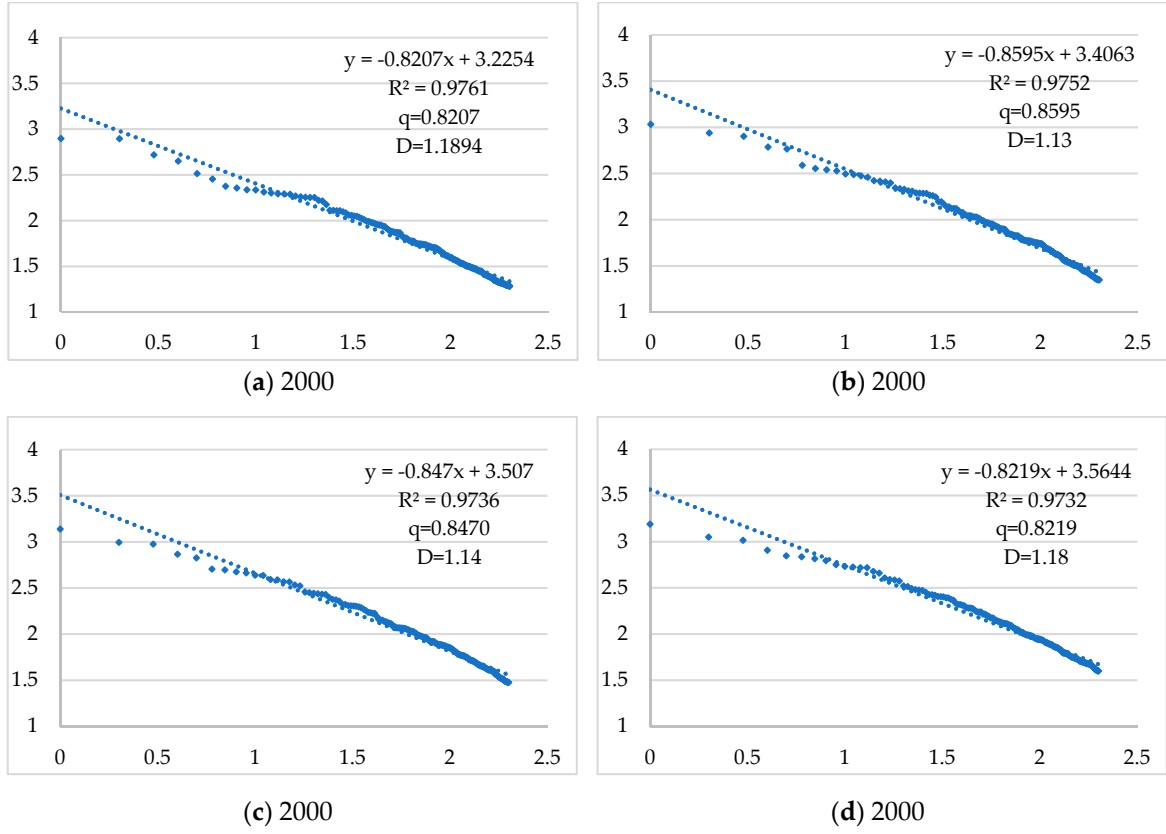

**Figure 7.** Cities' rank-size plots of China for 2000, 2005, 2010, 2015. The abscissa (x) is lg(R), where R stands for the city ranks of top200. The abscissa (x) is lg(R), where R stands for the city rank of top200. The vertical coordinates (y) is lg(S), where S stands for city size, namely urban area.

In 2000–2015, the gradual upward trend of the structural capacity over time indicates that the overall scale of Chinese cities continues to expand and the urban system is becoming increasingly complex. In the existing research on the evolution of urban land use scale distribution in China from 1990 to 2000, the scale distribution of urban land use in the country exhibits a decline in the Zipf index and a trend of increasing equilibrium. The research findings (Figure 8) from 2000 to 2015 indicate that the value of the Zipf index of China's urban system presented a rising trend and then decreased with time, from 0.82 in 2000 to a maximum of 0.86 in 2005, and then dropped to 0.82 in 2015. Additionally, the evolution of the city size distribution exhibited different phase characteristics in 2005.

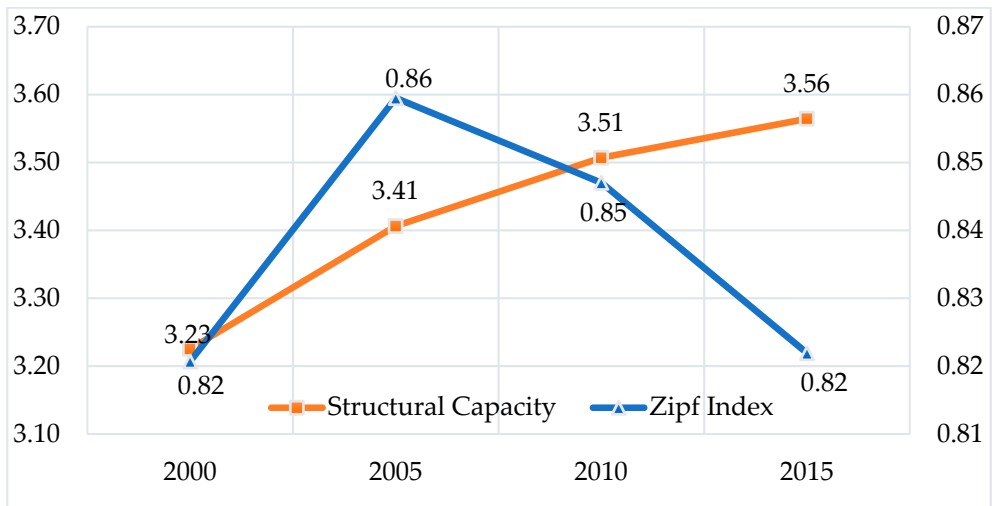

**Figure 8.** Change of Zipf Index and structural capacity in China's cities system from 2000–2015.

China implemented the urban development policy of "strictly controlling the scale of large cities and highlighting the development of small towns," combined with its urban development policy from 1996 to 2000, and implemented the diversified urban development policy of "coordinated development of large, medium, and small cities and towns" in 2000–2005 (Table 2).

**Table 2.** The policies and characteristics of urban development during different periods.

| Period | Policy | Characteristics |
|---|---|---|
| 2000–2005 | Strictly controlling the scale of large cities and highlighting the development of small towns | Rapid and extensive development of large cities |
| 2005–2010 | Intensive sustainable development of urbanization | Development of large cities is constrained |
| 2010–2015 | Taking the big cities as the basis, focusing on small and medium-sized cities, gradually forming a group of cities with large radiation effects, and promoting the coordinated development of large, medium, and small cities and towns | Coordinated development of large, medium, and small cities and towns |

Policy changes have led to the rapid and extensive development of large cities with natural advantages. The results of this study for 2000–2005 indicate that the expansion area of the top 10 large cities and megacities accounted for 31.50% of the total expansion area, and the expansion area of three megacities (area greater than 500 km²) accounted for a total expansion area of 12.48%. During this period, the top cities in China achieved remarkable development that considerably exceeded those of small and medium-sized cities. The first place rose, equilibrium declined, and the city size distribution was closer to the Czech Republic model.

In 2005, China proposed to follow the urbanization strategy of intensive development, which constrained the development of large cities to a certain extent. In the next 5 years, the proportion of

the top 10 cities expanded to 20.06%, and the top 3 cities were ranked. The proportion of urban areas decreased to 8.46%, the first degree of urban systems declined, equilibrium increased, and the Zipf index decreased.

After 2011, China adhered to the policy of "taking the big cities as the basis, focusing on small and medium-sized cities, gradually forming a group of cities with large radiation effects, and promoting the coordinated development of large, medium, and small cities and towns." The top 10 cities in 2000–2015 accounted for an expanded area. The ratio of the urban expansion area to the top 3 areas was further reduced to 5.80%. The first degree of the urban system and the Zipf index further declined, and the Chinese urban system became more balanced.

### 4.3. Analysis of Urban Expansion Characteristics

In 2015, the total urban area of prefecture-level cities reached 34,554.31 km$^2$, accounting for 2.17% of the total administrative area of the country and exhibiting an increase of 115.79% compared with that in 2000. The spatiotemporal expansion process of urban areas at the prefecture level and above is analyzed in four scales: cities, provinces, urban agglomerations, and economic zones. Figure 9 shows the detailed information concerning the spatiotemporal distribution characteristics of China's cities.

The analysis results indicate that among the top five cities in terms of urban area, Beijing, Shanghai, Tianjin, and Guangzhou occupied the top four places from 2000 to 2015. Wuhan ranked fifth in 2000, 2005, and 2010, but was replaced by Shenzhen in 2015. As the capital of China, Beijing ranks third in terms of urban area. It maintained an expansion intensity of 2.04% from 2000 to 2015, with a total expansion of 241.89 km$^2$, which is equivalent to one-third of the urban area in Beijing in 2000. Shanghai's urban area expanded more rapidly. In the past 15 years, its expansion strength of 6.44% expanded by 760.44 km$^2$, which is nearly double compared with its urban area of 787.31 km$^2$ in 2000. Tianjin has more than doubled in 15 years, and Guangzhou and Wuhan have expanded by 60%. The urban area of Chongqing in 2015 is nearly twice as large as it in 2000. In February 2010, the National Urban System Plan issued by the Ministry of Housing and Urban–Rural Development in Beijing identified Beijing, Tianjin, Guangzhou, and Chongqing as China's national central cities. At the national level, the aforementioned six cities are guaranteed to lead the cities in China. The collection and distribution functions can obtain the positioning of the national central city and is highly significant for the development of the region. The results of this study show that the central cities of the five major countries have expanded by 2,295.51 km$^2$ in 15 years, i.e., an increase of 82.45%. In addition, the five national central cities are the top five cities in the 2015 urban area rankings. The central cities of Chengdu, Wuhan, Zhengzhou, and Xi'an, which were identified later, ranked in the top 10, except for Zhengzhou. This result confirmed the practical guiding significance of the research results for the urban system planning of the country's comprehensive understanding of urban expansion.

The top three provinces with the largest urban area in 2015 were Jiangsu, Shandong, and Guangdong. The three provinces with the smallest urban areas were Tibet, Qinghai, and Hainan. The urban area of Hainan Province is small, but its urban area is high. In 2000–2015, the top three provinces with the largest urban expansion area and rate were Jiangsu, Shandong, and Guangdong. The three provinces with the smallest expansion area and speed were Tibet, Qinghai, and Hainan. The expansion rates of Jiangsu, which has the largest expansion area, and Tibet, which has the smallest expansion area, differ by more than 50 times. The top three provinces with the highest expansion intensity were Ningxia, Jiangxi, and Jiangsu, and their expansion strengths were 15.88%, 14.48%, and 13.32%, respectively. The latter three provinces and cities with smaller expansion intensity were Beijing, Henan, and Tibet, with expansion strengths of 2.04%, 3.04%, and 3.33%, respectively.

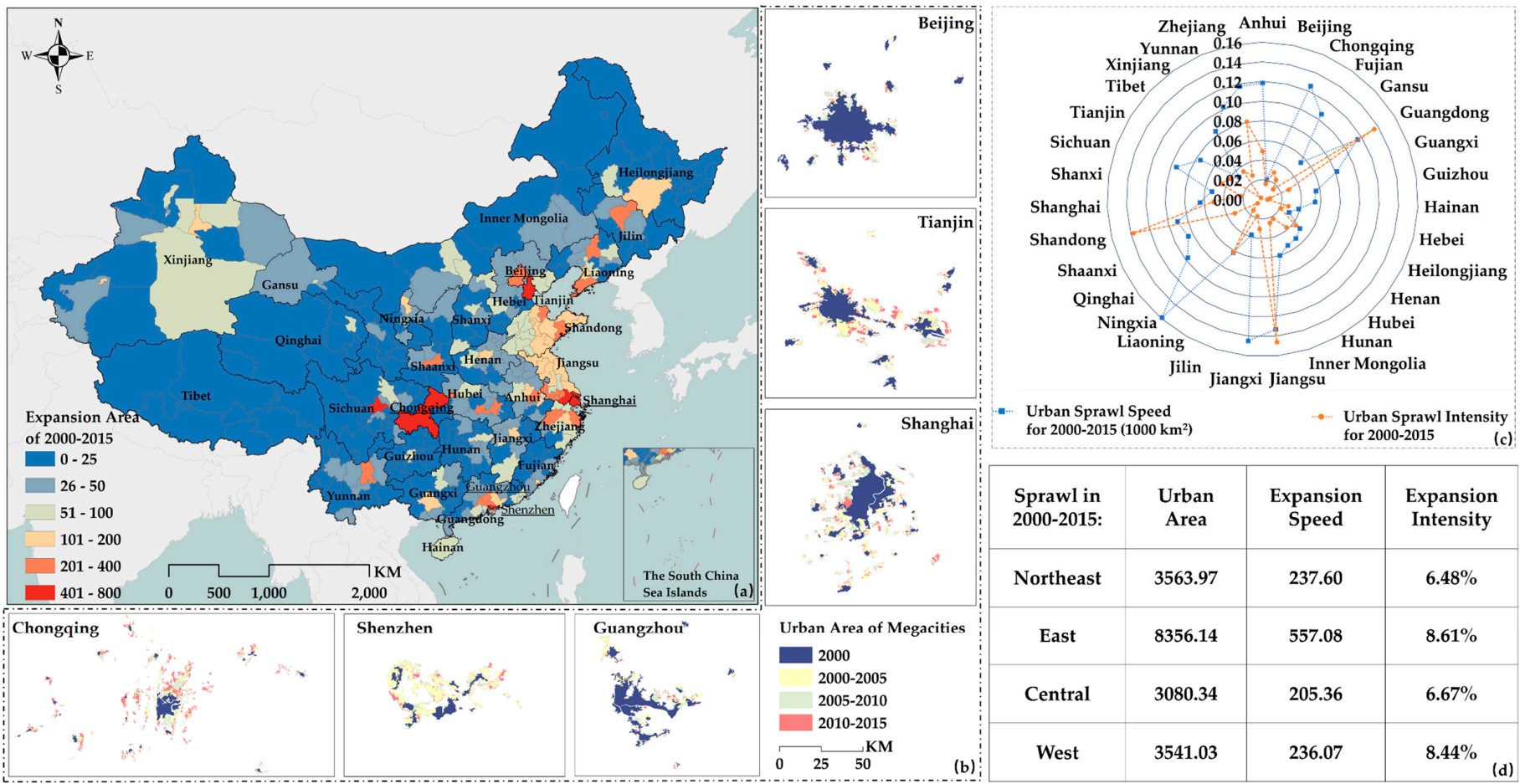

**Figure 9.** Spatiotemporal distribution map of urban expansion of China's cities in 2000–2015. Urban area and expansion area unit: km$^2$, Expansion speed unit: km$^2$ per year.

The urban areas of the four regions (Northeast China, East China, Central China, and West China) increased in stages based on the analysis of different economic zones. In 2015, the urban area of the eastern region was the largest, whereas that of the northeast region was the smallest. In terms of urban area, the largest in the eastern region was 3.79% in 2015 and the lowest in the western region was 0.92%, thereby indicating that the land urbanization level in the western region is still the lowest in the four regions of the country. From 2000 to 2015, the expansion area and rate of each region were ranked from high to low as follows: eastern region, western region, central region, and northeastern region. The extended area in the eastern region was 8356.14 km$^2$, and the expanded area in the northeastern region was 3563.97 km$^2$. From 2000 to 2015, the analysis of different economic zones showed that the expansion intensity of each region was ranked from high to low as follows: western region, eastern region, northeastern region, and central region. The expansion intensity in the eastern region reached 8.61%, whereas the expansion intensity in the central region was only 6.67%, which was the slowest growth in the area.

*4.4. Analysis of Urban Expansion Type*

The convex hull analysis method is used to determine and analyze the types of urban expansion in China. Figure 10a shows the expansion type of China's 31 provincial capitals. Figure 10b presents the specific percent of area in or out of the convex hull for China's provincial capital cities. From 2000 to 2015, China's capital cities of infilling type included Lanzhou, Guangzhou, Shanghai, Tianjin, Urumqi, Shijiazhuang, Guiyang, Hangzhou, Taiyuan, Jinan, Lhasa, and Wuhan. From 2000 to 2015, China's edge-expansion provincial capital cities included Harbin, Xi'an, Hefei, Yinchuan, Nanning, Kunming, Fuzhou, Nanchang, Nanjing, Chongqing, Changsha, Xining, Zhengzhou, Chengdu, Hohhot, Shenyang, Haikou, Beijing, and Changchun. Harbin is the most extensive city and Lanzhou is the most intensive one. Some megacities, such as Guangzhou, Tianjin and Shanghai, also lead the way in terms of intensive development. But, the urban areas of Chongqing are widely dispersed geographically.

From 2000 to 2015, the number of infilling type cities accounted for 26.7% and included Taiyuan City, Jiamusi City, Yichang City, Baishan City, Panjin City, Yanbian Korean Autonomous Prefecture, Shiyan City, Liaoyang City, Lanzhou City, Zaozhuang City, Datong City, and other 90 cities, including Jinzhou City, Baoji City, and Luoyang City. The proportion of extended cities accounted for 73.3% and included Turpan, Fuyang, Bortala Mongolian Autonomous Prefecture, Beijing, Tianjin, Huizhou, Fangchenggang, Wuzhong, Lu'an, Hebi, Qingyuan, Xiamen, and 247 cities, including Zhongwei City. From 2000 to 2015, the expansion of urban space at the prefecture level and above was primarily epitaxial, the urban form tended to be scattered, and urban land use efficiency declined. Moreover, all of the cities in edge-expansion type are Fujian, Hainan, Jiangsu, Ningxia and Anhui. The most intensive province is Jilin and Tibet, Shandong, Shanxi and Heilongjiang are also more intensive than extensive. And 61.11% of the cities in Northeast China are infilling type, 31.33% for Central China, 19.39% for East China and 18.80% for West China.

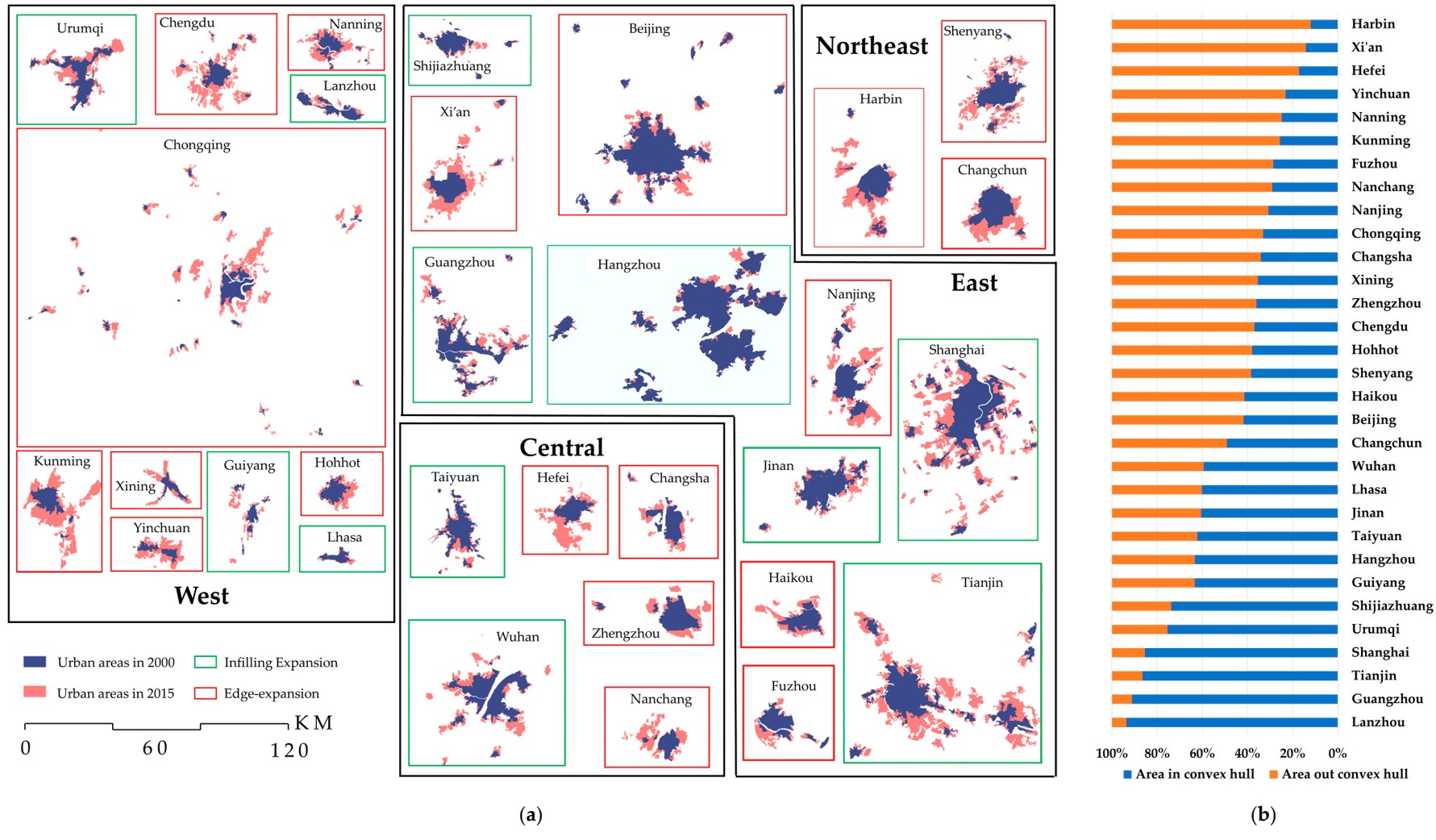

**Figure 10.** Expansion types of capital cities in China from 2000 to 2015.

### 4.5. Analysis of Land Use Change in Urban Sprawl Areas

From 2000 to 2015, 11,106.45 km$^2$ cultivated land, 1906.27 km$^2$ forest and grassland, and 1992.55 km$^2$ water area and unutilized land have been converted into urban areas (Figure 11). Urbanization has destroyed forest and grassland to the extent double the size of the Beijing urban area. About 10% of the sprawl area is at the cost of forest and grassland. More than half of the sprawl area consists of cultivated land; water areas and unutilized land have also suffered great losses. Urbanization of China is at the expense of nature and arable land. It seriously threatens people's livelihoods and biodiversity in the urban landscape.

The analysis was carried out with province as the unit. Urban sprawl in 28 provinces, except for Beijing, Tibet and Guangzhou, is dominated by the occupation of cultivated land. The expansion of Beijing in this period is combined construction land such as rural residential areas and occupied cultivated land. The urban expansion of Tibet is mainly from forest and grass land and Guangdong is a water area and unused land.

From 2000 to 2015, Zhejiang, Chongqing, Anhui, Henan, Jilin, Jiangsu, Shanghai, Ningxia, Shandong and Shanxi rank the top 10 provinces for the area of cultivated land occupied by urban expansion. Tibet, Hunan, Hainan, Xinjiang, Inner Mongolia, Guizhou, Jiangxi, Yunnan, Guangxi and Fujian are the top 10 provinces in terms of land area occupied by forest and grass. The top 10 provinces in terms of occupied construction land area are Beijing, Shaanxi, Sichuan, Guangxi, Hebei, Qinghai, Hainan, Liaoning, Fujian And Shanxi. The top 10 provinces with occupied water area and unutilized land area are: Guangdong, Gansu, Tianjin, Hubei, Guizhou, Xinjiang, Heilongjiang, Fujian, Ningxia and Inner Mongolia.

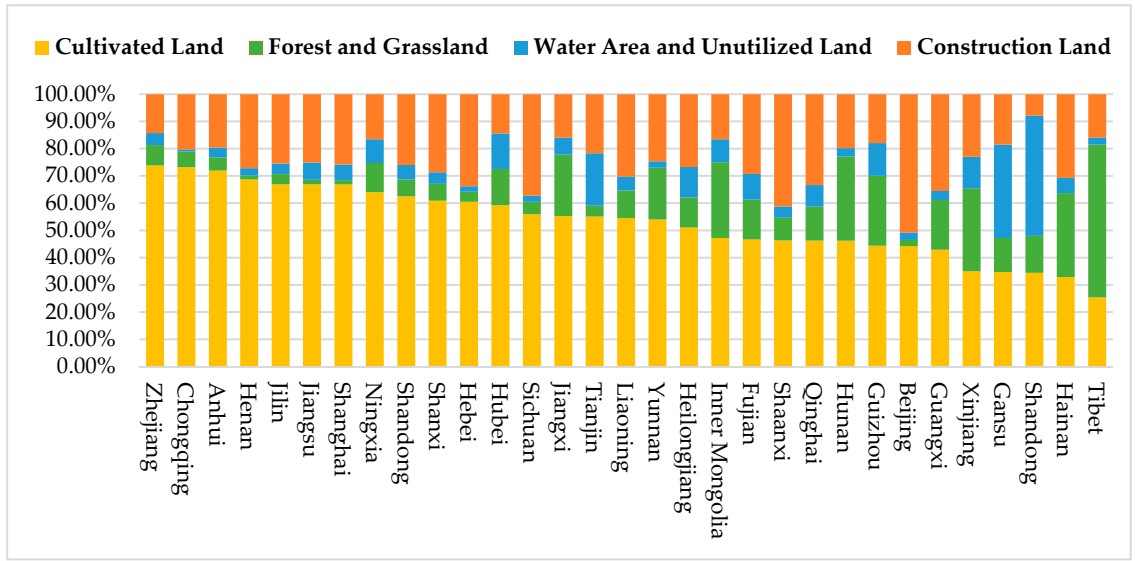

**Figure 11.** Proportion of land use occupied by urban sprawl areas of China's cities 2000–2015.

## 5. Conclusions

This study proposes a semi-automatic method for urban area extraction based on high-resolution remote sensing imagery and geographic information data-assisted automatic computer recognition and manual interpretation. It has obtained 337 prefecture-level cities in China in 2000, 2005, and 2010. The conclusions based on the high-precision urban area of the fourth phase (2015) and the analyses of urban spatiotemporal patterns are as follows:

(1) From 2000 to 2015, China's cities generally expanded rapidly and maintained an exponential growth trend. By 2015, the urban area was equivalent to 2.16 times that in 2000. The urban areas are mostly concentrated in the eastern and central regions, and the eastern and western regions are considerably different. However, the western region is continuously expanding.

(2) The overall scale of Chinese cities continues to expand, the urban system is becoming increasingly complex, the scale distribution is more dispersed than the ideal Zipf distribution, the scale of high-level cities is insufficiently prominent, and small and medium-sized cities are relatively developed. After 2005, the urban system has become increasingly balanced.

(3) The urban areas of Beijing, Shanghai, Tianjin, and Guangzhou always occupied the top four places in 2000–2015. In 2015, Shenzhen ranked fourth and Chongqing ranked sixth. Five national central cities and China's four major lines have been identified. The cities' positioning remains the same.

(4) In the study area, 73.30% of the cities exhibit expansive expansion, the urban form tends to be scattered, and land use efficiency is low.

(5) The new urban areas mainly come from cultivated land and ecological land (forest, grassland, water area); the urbanization threatens people's livelihoods and biodiversity.

The proposed method needs high-resolution imagery and high-quality thematic data to ensure accuracy; however, the storage and management of so much data is a significant problem. This study only analyzes three aspects: urban size distribution, extended regional characteristics, and expansion types. The next step will be to combine urban, economic, population, land use, and other thematic data to investigate urban expansion coordination, land use efficiency, spatial form change, and land occupation type. A comprehensive analysis of other aspects will be conducted to explore other urban development rules in China.

**Author Contributions:** Hanchao Zhang conceived and designed the research; Hao Wang and Xiaogang Ning contributed materials; and Hanchao Zhang and Zhenfeng Shao wrote the paper.

**Funding:** This work was supported in part by the National key R & D plan on strategic international scientific and technological innovation cooperation special project under Grant 2016YFE0202300, the National Natural Science Foundation of China under Grants 61671332, 41771452, 41401513, and 41771454, the Natural Science Fund of Hubei Province in China under Grant 2018CFA007, Fundamental Research Funds for the Central Public-interest Scientific Institution under Grant 7771803, Beijing Key Laboratory of Urban Spatial Information Engineering under Grant 2018202.

**Acknowledgments:** This work was supported by National 973 Basic Research and Development Program project (No. 2012CB719904), The National Key Research and Development Program of China (No. 2016YFE0205300), National Natural Science Foundation of China (No. 41401513), National Science and Technology Specific Projects (No. 2012YQ1601850 and No. 2013BAH42F03), Program for New Century Excellent Talents in University (No. NCET-12-0426), Beijing Key Laboratory of Urban Spatial Information Engineering (No. 2018202).

**Conflicts of Interest:** The authors declare no conflict of interest.

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
