# Peer review of "Spatiotemporal Pattern Analysis of China’s Cities Based on High-Resolution Imagery from 2000 to 2015"

_ijgi, doi:10.3390/ijgi8050241_

Round 1
Reviewer 1 Report
Spatiotemporal Pattern Analysis of China's Cities Spatiotemporal Pattern Analysis of China's Cities
The authors used very high-resolution remote sensing data and image classification tools (semi-automatic) to examine the urban land-use change in major cities of China from 2000 to 2015. Results confirm anecdotal assertions and facts observed by other authors that the growth trend of Chinese cities has continued at a high-speed from 2000 to 2015. They discussed how gradual expansion of the cities will result in increasing complexity of Chinese urban structures.
Recommendation
I think this is an interesting paper that is important because it looks at using very high-resolution satellite images to map and monitor urban landuse changes systematically using advanced analytical and remote-sensing information extraction methods. I think the analysis was strong and I found the results convincing. However, beyond extracting the time stamps of urban footprint or boundaries, it would be useful to include other interesting aspects of urban expansion science such as drivers of the change and implication on biodiversity and urban microclimate change.
General questions
1. Urban forests are a significant social and economic resource for urban residents (urban ecology and nature, food, medicine, aesthetics, cultural value and human wellbeing). Urban forest loss, therefore, threatens people’s livelihoods as well as biodiversity in the urban landscape. A brief discussion of the consequences of urban expansion is not mentioned in the manuscript and should be discussed. Particularly since multitemporal images of the cities were used to monitor changes in urban extent, such expansion threatens natural landscapes, which are valuable in maintaining rich urban biodiversity.
2. The authors did not mention if the semi-automatic approach is rule-based, which needs to be constantly crafted and can be challenging on its own.
3. It is innovative to develop or explore new information extraction techniques to efficiently extract urban footprint from very-high-resolution satellite imageries as the authors attempted (i.e. conceptualization, development and implementation of the new method as they claim). However, the authors did not consider the transferability of the semi-automatic method developed on very-high-resolution satellite images of similar or slightly different regions to test the robustness of the newly developed approach and potential applications.
4. The authors did not also provide evidence of comparison of the new methods with other existing approaches to address questions inconsistency in concepts and extraction standards, low precision, and poor comparability existing in urban monitoring which may lead to the wrong conclusion as opined.
Comments by line number
Line 137: The table does not include the month images were collected?
Line 148: The figure is not visible.
Line 229: Figure 3 (a) depicts building instead of high rise or tall buildings, since (b) shows low-rise buildings.
Author Response
Response to Reviewer 1 Comments
Point 1: Urban forests are a significant social and economic resource for urban residents (urban ecology and nature, food, medicine, aesthetics, cultural value and human wellbeing). Urban forest loss, therefore, threatens people’s livelihoods as well as biodiversity in the urban landscape. A brief discussion of the consequences of urban expansion is not mentioned in the manuscript and should be discussed. Particularly since multitemporal images of the cities were used to monitor changes in urban extent, such expansion threatens natural landscapes, which are valuable in maintaining rich urban biodiversity.
Response 1: Thank you for your suggestion and we accept it. Analysis of land use change and a discussion of the consequences of urban expansion are added in Chapter 5. We find that the urbanization has destroyed forest and grassland two size of Beijing urban area. About 10% of the sprawl area is at the cost of forest and grassland. More than half of the sprawl area consists of cultivated land. And water area and unutilized land also suffer great losses. Urbanization of China is at the expense of nature and arable land. It seriously threatens people’s livelihoods and biodiversity in the urban landscape.
Point 2: The authors did not mention if the semi-automatic approach is rule-based, which needs to be constantly crafted and can be challenging on its own.
Response 2: Thank you. The automatic part is based on features and the manual part is based on rule.
First, a method fusing right-angled and right-angled features which we proposed in 2017[1] is used to automatic extract residential polygons from high-resolution remote sensing images. The correctness, completeness, and quality of this method is higher 6.76%,10.12%,12.14% respectively than the existed method. The automatic method uses feature-level-based fusion of right-angle-corners and right-angle-sides to extract the human settlements. It is composed of five steps, namely (1) detection of line segments, (2) detection of Harris corners, (3) verification of corners by line segments, (4) construction of built-up area index, (5) thresholding of human-settlement index.
Then, the polygons of the settlements are superimposed with the location of the district government, whereas the polygons of the concentrated contiguous settlements, where the district government is located, are used as the initial urban area. The classification system and standards for geographic country monitoring indicate that the original urban area should be manually interpreted and modified in accordance with the urban area extraction principles and rules to obtain the final urban area.
The automatic part has been testing in [1]. The manual part has been examined quality inspection station of each province and the result has been applied to many government departments.
[1] Lin, X.; Ning, X. Extraction of human settlements from high resolution remote sensing imagery by fusing features of right angle corners and right angle sides. Acta Geodaetica Et Cartographica Sinica 2017, 46, 83-89. https://www.int-arch-photogramm-remote-sens-spatial-inf-sci.net/XLII-2-W7/803/2017/isprs-archives-XLII-2-W7-803-2017.pdf
Point 3: It is innovative to develop or explore new information extraction techniques to efficiently extract urban footprint from very-high-resolution satellite imageries as the authors attempted (i.e. conceptualization, development and implementation of the new method as they claim). However, the authors did not consider the transferability of the semi-automatic method developed on very-high-resolution satellite images of similar or slightly different regions to test the robustness of the newly developed approach and potential applications.
Response 3: Thank you for your question. The test of robustness in different regions must be done before extracting of the 337 cities' urban. These so many cities have their own characteristics and are more or less different from each other. The keys of obtaining a standardized result are the principles which are summarized from the rich experience of many different cities' urban extraction using different satellite images in a long time. And the result is examined by professional inspectors and users. So, we think our method is robust and can be tested.
Point 4: The authors did not also provide evidence of comparison of the new methods with other existing approaches to address questions inconsistency in concepts and extraction standards, low precision, and poor comparability existing in urban monitoring which may lead to the wrong conclusion as opined.
Response 4: Thank you for your suggestion and we have compared the proposed method with some existing approaches in Chapter 4. The result is compared with other results such as the result of MODIS in the report ”East Asia's Changing Urban Landscape” and the result of manual extraction by Landsat images.
Point 5: The table does not include the month images were collected?
Response 5: Thank you for your question. The months of images are introduced above the table, Satellite imagery was mainly obtained from September to December and small number (about 5%) of imagery is obtained in other months.
Point 6: The figure is not visible.
Response 6: I'm so sorry for that. We have added the Figure 2 by .jpg rather than .vsdx so that the online system can correctly transfer a docx to a pdf.
Point 7: Figure 3 (a) depicts building instead of high rise or tall buildings, since (b) shows low-rise buildings.
Response 7: Thank you for your suggestion and we accept it.

Reviewer 2 Report
The paper is of interest and could be worthy of publishing, once some explanation of methods and results (as noted above) are addressed.
I have scanned back a copy of the manuscript on which I have made some editorial suggestions as well as ways in which I feel the text and supporting figures and/or tables could be improved. There are several sections in which I think the authors should re-phrase so that it is crystal clear you are doing a temporal study.
Overall suggestions:
Figure 1: need to better distinguish Northeast China and East China;
re: line 107 (and elsewhere where you've discussed the selected 337 cities under study). You have not always been consistent in describing them (here they include prefecture-level cities, but elsewhere only above-preferecture). Might also be helpful discuss here the population ranges of included cities, since your conclusion highlights the rapid growth rates in some of China's megacities;
Table 1: as my handwritten comments suggest, the table content needs to be augmented to include other data parameters (such as resolution) and a better alignment on its content and accompanying text;
Figure 2 was not visible to the reviewers in the pdf provided;
The principles for "Urban Area Extraction" (page 6) were overall helpful; principle (4) for determining urban landscapes needs to be clarified … is this for determining "built up" versus non built-up areas; this principle seems to include residential (that is, housing) component but not the commercial or institutional portions of a city -- not sure how you can readily resolve this using the scales you are at;
Figure 3: more information needed in the caption as noted by handwritten comments;
re Line 233, some additional citations (or references) are needed to substantiate the frequent use of these indicators;
re: Line 261, mention of a "structural capacity (Sc)' term which is lnP1...later in the manuscript, there is some confusion noted (see later comments);
re: Line 291 (again some confusion noted in handwritten comments confusing how the 337 study cities are described);
re: Line 304 (how is construction land treated in the final analysis, since it on the way to something else)
re: Line 316 the term 'exponential' implies run away growth and has a specific scientific/mathematical meaning. Growth in these cities is significant but NOT exponential;
Figure 4: several comments made on this figure, particularly some confusion on the structural capacity term presented here versus that on Line 261; some clarity needed to clearly label the figure so trends are visble;
re: Page 11 (lines 342 to 361), feel this paragraph needs a data table and the paragraph needs to be re-written to clear explain what constitutes a megacity, the change(s) in government policies that have influenced them over time and what your results actually show; the data table should include the changing characteristics noted in the mega cities, as the changes in mega cities seems a key conclusion in the paper;
Figure 6 (page 12) needs to be re-thought. I think the map would be better as a choropleth map showing urban expansion rates by prefecture. The two circled elements (i.e. speed and intensity of sprawl and urban sprawl speed ) could be separate diagrams as they are too small in print to see what is shown; a more thorough discussion is needed throughout this section to explain Area Speed and Intensity statistics shown;
Figure 7 (discussed in text on pages 13-14) would be better as a table -- I would include a map of two example cities (perhaps one showing the infilling character and one showing the edge expansion character)

Author Response
Response to Reviewer 2 Comments
Point 1: Figure 1: need to better distinguish Northeast China and East China.
Response 1: We accept the suggestion and have changed the symbol.
Point 2: line 107 (and elsewhere where you've discussed the selected 337 cities under study). You have not always been consistent in describing them (here they include prefecture-level cities, but elsewhere only above-preferecture). Might also be helpful discuss here the population ranges of included cities, since your conclusion highlights the rapid growth rates in some of China's megacities.
Response 2: Thank you for your suggestion and we accept it. Some information about the population ranges of these cities are add to Section 2.1.
Point 3: Table 1: as my handwritten comments suggest, the table content needs to be augmented to include other data parameters (such as resolution) and a better alignment on its content and accompanying text.
Response 3: We accept this suggestion. The resolution, months of the data are added and a better alignment has been made.
Point 4: Figure 2 was not visible to the reviewers in the pdf provided.
Response 4: I'm so sorry for that. We have added the Figure 2 by .jpg rather than .vsdx so that the online system can correctly transfer a docx to a pdf.
``
Point 5: The principles for "Urban Area Extraction" (page 6) were overall helpful; principle (4) for determining urban landscapes needs to be clarified … is this for determining "built up" versus non built-up areas; this principle seems to include residential (that is, housing) component but not the commercial or institutional portions of a city -- not sure how you can readily resolve this using the scales you are at.
Response 5: Thank you for your question. Principle (4) is about how to judge a region considering its surroundings. Of course, the "built up" areas includes the commercial or institutional portions. Meanwhile, these areas like commercial or institutional portions are not easy to identify only by remote sensing imagery since they are all buildings there, and the data of Geographical Conditions Monitoring Project can help to make it clear when we meet difficult situations. In fact, this kind of situations are not so many, a cluster of high-rise buildings among other clusters of high-rise buildings must be urban whatever they are institutional, commercial or resident. The regions at the edge of urban also can be judged by surroundings and their characteristics, mostly. So, the place and its surroundings make the urban landscapes, and the urban landscapes are important to make the place urban or not urban.
Point 6: Figure 3: more information needed in the caption as noted by handwritten comments.
Response 6: We accept this suggestion and more information has been added.
Point 7: Line 233, some additional citations (or references) are needed to substantiate the frequent use of these indicators.
Response 7: Thank you for your suggestion and we have added some citations.
Point 8: Line 261, mention of a "structural capacity (Sc)' term which is lnP1...later in the manuscript, there is some confusion noted (see later comments).
Response 8: Thank you for your suggestion. We have checked and corrected this part. LnP1 should be lgP1.
Point 9: Line 291 (again some confusion noted in handwritten comments confusing how the 337 study cities are described).
Response 9: Thank you. These 337 cities are cities at prefecture level and above including capital cities, prefecture-level cities, autonomous prefectures, regions, and alliances in mainland China. This has been corrected.
Point 10: Line 304 (how is construction land treated in the final analysis, since it on the way to something else).
Response 10: Thank you for your question. Forgive me for my poor English, I meant to express that the construction land can be separated using high-resolution remote sensing images and difficult to be separated by MODIS or Landsat images. We have corrected this.
Point 11: Line 316 the term 'exponential' implies run away growth and has a specific scientific/mathematical meaning. Growth in these cities is significant but NOT exponential.
Response 11: Thank you for your suggestion and we have corrected it.
Point 12: Figure 4: several comments made on this figure, particularly some confusion on the structural capacity term presented here versus that on Line 261; some clarity needed to clearly label the figure so trends are visible.
Response 12: We accept your kind suggestions and make changes in the appropriate location of Figure 4.
Point 13: Page 11 (lines 342 to 361), feel this paragraph needs a data table and the paragraph needs to be re-written to clear explain what constitutes a megacity, the change(s) in government policies that have influenced them over time and what your results actually show; the data table should include the changing characteristics noted in the mega cities, as the changes in mega cities seems a key conclusion in the paper.
Response 13: We accept your suggestion and re-write this paragraph so that it can be clearer for reader to understand.
Point 14: Figure 6 (page 12) needs to be re-thought. I think the map would be better as a choropleth map showing urban expansion rates by prefecture. The two circled elements (i.e. speed and intensity of sprawl and urban sprawl speed ) could be separate diagrams as they are too small in print to see what is shown; a more thorough discussion is needed throughout this section to explain Area Speed and Intensity statistics shown.
Response 14: Thank you for your suggestion and we have update Figure 6 and this paragraph.
Point 15: Figure 7 (discussed in text on pages 13-14) would be better as a table -- I would include a map of two example cities (perhaps one showing the infilling character and one showing the edge expansion character).
Response 15: We accept your suggestion and two example cites have been added.

Reviewer 3 Report
It is well known that China in the past 20 years has grown rapidly, that the definition of urban has changed and that Chinese planning authorities have changed policy goals at intervals. This study tries to use satellite imagery, auAi algorithms and fractal systems to produce an accurate assessment of the results. The basic picture produced is fascinating and exciting as Shenzhen, for instance, replaces Xian in rank order. But the introduction of Zipf’s law,an American statistical anomaly of its century of growth, seems arbitrary and extraneous to the research and conclusions found here. The hierarchy and distribution of cities in China might once have followed this law, but with centuries of planning and the current planning institutes and design offices it is hardly surprising that The Chinese pattern is so different. It would help a great deal if the structure and hierarchy of Chinese cities was discussed further. Also the definition of urbanization boundary and actual urban build out is unclear and important, making the definition of urbanization rate and extent in some vast city territories in western China unclear. This paper contained important insights but needs further clarification.
Author Response
Response to Reviewer 3 Comments
Point 1: It is well known that China in the past 20 years has grown rapidly, that the definition of urban has changed and that Chinese planning authorities have changed policy goals at intervals. This study tries to use satellite imagery, auAi algorithms and fractal systems to produce an accurate assessment of the results. The basic picture produced is fascinating and exciting as Shenzhen, for instance, replaces Xian in rank order. But the introduction of Zipf’s law,an American statistical anomaly of its century of growth, seems arbitrary and extraneous to the research and conclusions found here. The hierarchy and distribution of cities in China might once have followed this law, but with centuries of planning and the current planning institutes and design offices it is hardly surprising that The Chinese pattern is so different. It would help a great deal if the structure and hierarchy of Chinese cities was discussed further. Also the definition of urbanization boundary and actual urban build out is unclear and important, making the definition of urbanization rate and extent in some vast city territories in western China unclear. This paper contained important insights but needs further clarification.
.
Response 1: We are very grateful for your questions and suggestions.
(1) Zipf's law is proposed by George Kingsley Zip in word frequency statistics, 1949 and then applied in assessment of American and other country's cities' distribution. It is not just a law to obey but a statistical technique to analyse the structure and hierarchy of cities. The rank-size rule has been a quantitative theory and method in the study of urban geography.
Cities' distribution analysis is important for giving a brief outlook to understand the structure and hierarchy for a country's cities. The rank-size rule and Zipf's law are not only for natural cities but also a useful tool for analysing the influence of government policies. So, we think it is necessary and important to the research and conclusions. And we accept your suggestion and discuss the structure and hierarchy of Chinese cities further.
And many researchers have used rank-size law to analyse the city distribution. Jeff Luckstead[1] on China's and India's urban rank-size analysis when the population is more than 750000 city as the research samples, Fang C[2] and Cheng K M[3] selected capital city and cities at or above the prefecture level as the research samples to analyse Zipf index. Chen Z[4] first verifies the scale-order distribution of Chinese cities, and then divides them into tourism cities, provincial capital cities, coastal cities, Yangtze river delta cities, pearl river delta cities, southwest cities and northeast cities, and conducts parallel growth analysis.
All in all, we think the analysis of rank-size rule is important and we have done more research.
(2) We have given more details of the definition of urbanization boundary and actual urban build out in Section 3.2.
[1] Luckstead J, Devadoss S. A comparison of city size distributions for China and India from 1950 to 2010[J]. Economics Letters, 2014, 124(2):290-295.
[2] Fang C, Wang Z. Quantitative Diagnoses and Comprehensive Evaluations of the Rationality of Chinese Urban Development Patterns[J]. Sustainability, 2015, 7(4):3859-3884.
[3] Cheng K M, Zhuang Y J. Spatial Econometric Analysis of The Rank-size Rule for Urban System:A Case of Prefectural-level cities in China’s Middle Area[J]. Scientia Geographica Sinica, 2
[4] Chen Z, Fu S, Zhang D. Searching for the Parallel Growth of Cities[J]. Urban Studies, 2013, 50(10):2118-2135.

Reviewer 4 Report
The study aim is to present the Spatiotemporal Pattern Analysis of China's Cities Based on High-resolution Imagery from 2000 to 2015. The manuscript is presented in a clear and nice way. However to have scientific merit the following comments should be addressed.
1. Need to add citations for line 33 to 43, 49 to 52
2. Better to add some sample studies for line 68 to 76
3. Figure 1 – better to use a different symbol for capital and other cities
4. Figure 2 is missing, then, how to understand the flowchart of the study
5. The method of mosaicking is missing. How did you mosaic several spatial resolution data such as Landsat, Ikonos, QuickBird
6. Need to explain more about the urban area extraction by using manual and automatic extraction method. Also, need to explain the limitation of the used method.
7. How did you deal with shadow effects- need to explain?
8. The discussion mainly based on the result of this study. You have to clarify your result by using previous studies which were published by other scholars. It can be used as some validation process.
9. You have to explain the limitation of the study.
Author Response
Response to Reviewer 4 Comments
Point 1: Need to add citations for line 33 to 43, 49 to 52.
Response 1: Thank you for your suggestion. We have added some citation for line 33 to 43, 49 to 52.
Point 2: Better to add some sample studies for line 68 to 76.
Response 2: Thank you for your suggestion. We have added some sample studies for line 68 to 76.
Point 3: Figure 1 – better to use a different symbol for capital and other cities.
Response 3: Thank you for your suggestion and we have replaced the symbol by a star symbol for capitals.
Point 4: Figure 2 is missing, then, how to understand the flowchart of the study.
Response 4: I'm so sorry for that. We have added the Figure 2 by .jpg rather than .vsdx so that the online system can correctly transfer a docx to a pdf.
Point 5: The method of mosaicking is missing. How did you mosaic several spatial resolution data such as Landsat, Ikonos, QuickBird.
Response 5: Thank you for you question. We have added some details for mosaicking in order to be clearer for readers.
Point 6: Need to explain more about the urban area extraction by using manual and automatic extraction method. Also, need to explain the limitation of the used method.
Response 6: We are grateful for your suggestion. More explanation about the urban area extraction method has been added. And the limitation is discussed in 4.1.
Point 7: How did you deal with shadow effects- need to explain?
Response 7: Thank you for your question. Yes, shadow effects are big problems for auto interpretation and we use the traditional manual visual interpretation way to find out the boundary of urban in the shadows.
Point 8: The discussion mainly based on the result of this study. You have to clarify your result by using previous studies which were published by other scholars. It can be used as some validation process.
Response 8: We accept this suggestion. Some previous studies have been brought in for comparing.
Point 9: You have to explain the limitation of the study.
Response 9: We accept this suggestion. The limitation of the study has been explained at Chapter 5.

Round 2
Reviewer 4 Report
Thanks to the authors for their great efforts to incorporate my comments into the manuscript. Most of the comments have been considered and, in my view, the quality of the article has improved considerably. I can accept this article for the publication. I wish all success for future research.